#### Evaluation of Pandora HCHO and NO2 with Airborne In Situ Observations 1

- 2 Abby Sebol<sup>1</sup>, Glenn M. Wolfe<sup>2</sup>, Timothy Canty<sup>1</sup>, Jason St. Clair<sup>2,3</sup>, Erin Delaria<sup>2</sup>, Jennifer Kaiser<sup>4</sup>, Nidhi
- Desai<sup>4</sup>, Andrew Rollins<sup>5</sup>, Eleanor Waxman<sup>5,6</sup>, Kristen Zuraski<sup>5,6</sup>, Bryan Place<sup>2,7</sup>, Apoorva Pandey<sup>2,3</sup>, 3
- Akanksha Singh<sup>1</sup>, Allison Ring<sup>1</sup>, Charles Gatebe<sup>8</sup>, Jonathan Dean-Day<sup>9</sup> 4
- <sup>1</sup>Department of Atmospheric and Oceanic Science, University of Maryland, College Park, MD, USA 5
- <sup>2</sup>Atmospheric Chemistry and Dynamics Laboratory, NASA Goddard Space Flight Center Greenbelt, MD, USA
- <sup>3</sup>University of Maryland, Baltimore County, Baltimore, MD, USA
- <sup>4</sup>School of Earth and Atmospheric Sciences, Georgia Institute of Technology, Atlanta, GA, USA
- <sup>5</sup>NOAA Chemical Science Laboratory, Boulder, CO, USA 9
- <sup>6</sup>University of Colorado Boulder, CIRES, Boulder, CO, USA
- <sup>7</sup>Sciglob Instruments and Services LLC, Columbia, MD, USA
- <sup>8</sup>NASA Ames Research Center, Moffett Field, CA, USA
- 13 <sup>9</sup>Bay Area Environmental Research Institute, Moffett Field, CA, USA
- Correspondence to: Abby Sebol (asebol@umd.edu)

16

21

29

Abstract: The Pandora Global Network (PGN) is a system of ground-based spectrometers reporting continuous daytime column HCHO and NO2. While the Direct Sun (DS) NO2 product has been well studied, the Multi Axis Differential Optical Absorption Spectroscopy (MAX DOAS) products are largely unvalidated. Using the Atmospheric Emissions and Reactions Observed from Megacities to Marine Areas (AEROMMA) airborne campaign in the summer of 2023, we evaluate the performance of select Pandora monitors relative to in situ airborne observations. A case study over a Pandora in the California desert shows MAX DOAS HCHO captures the total tropospheric column (within 4% of the integrated in situ column) but does not match the vertical shape of the HCHO profile where the Pandora is biased high near the surface and low near the top of the boundary layer. The MAX DOAS NO2 is 80% lower for the entire profile and is particularly sensitive to the viewing angle of the Pandora due to the spatial heterogeneity of NO<sub>2</sub>. Ten Pandoras located in the New York City (NYC) domain capture the day to day variability of HCHO as well as spatial gradients from New Jersey to NYC to Long Island. The mean NYC Pandora HCHO correlates well with mean Tropospheric Emissions Monitoring of Pollution (TEMPO) HCHO columns of a similar domain on clear sky days. On those days, MAX DOAS columns exhibit a lower slope (slope = 0.78, y-intercept= $1.08 \times 10^{15} \text{ molec/cm}^2$ ;  $R^2 = 0.62$ ) while DS columns show a higher offset (slope = 0.90, y-intercept=2.83 x 10<sup>15</sup> molec/cm<sup>2</sup>; R<sup>2</sup> = 0.63). These results demonstrate the value of Pandora HCHO products while highlighting the need for improved uncertainty quantification.

#### 1. Introduction

The Pandora Global Network (PGN) is comprised of ground-based UV-VIS spectrometers that report total and partial column formaldehyde (HCHO), nitrogen dioxide (NO<sub>2</sub>), and ozone at over 150 operational monitors across six continents (<a href="https://pandora.gsfc.nasa.gov/PGN/">https://pandora.gsfc.nasa.gov/PGN/</a>; Last access July 2025). Pandoras operate in two modes: Direct Sun (DS) and Multi Axis Differential Optical Absorption Spectroscopy (MAX DOAS). In the DS mode, the Pandora sensor follows the sun as it moves across the sky and column amounts are derived using differential absorption. The MAX DOAS mode measures scattered light at different elevation angles to calculate a tropospheric column, near-surface partial columns, and a surface concentration. Understanding ozone and its precursors such as NO<sub>2</sub> and HCHO is an important priority for the U.S Environmental Protection Agency (EPA) under the NAAQS final rule in 2015 (EPA, 2015).

First established in 2006, PGN released its latest data version in 2020 after numerous hardware and software updates. DS NO<sub>2</sub> is likely to have less uncertainty than the MAX DOAS NO<sub>2</sub> retrievals due to simplified path integration (Cede et al., 2006; Herman et al., 2009). Total NO<sub>2</sub> columns agree well with satellite retrievals such as the Ozone Monitoring Instrument and the Tropospheric Monitoring Instrument (Herman et al., 2009; Herman et al., 2018; Kotsakis et al., 2022) and airborne remote sensors (Judd et al., 2019; Judd et al., 2020; Tzortziou et al., 2023). However, each Pandora spectrometer is calibrated and operated independently, leading to variability in performance across the network (Szykman et al., 2025; Rawat et al., 2025).

The updated HCHO products (Spinei et al., 2021) are relatively new with data beginning around 2020. In contrast to NO<sub>2</sub>, the DS HCHO is likely to have more uncertainty than the MAX DOAS HCHO because of instrumentation drift (Cede, 2025). Validation studies are crucial to improving confidence in Pandora products, especially as the PGN is a key component of the validation plan of Tropospheric Emissions Monitoring of Pollution (TEMPO), the data record for which begins August 2023 (Szykman, 2023).

Here, we evaluate Pandora retrievals against in-situ airborne observations collected in Southern California (CA), New Jersey (NJ), New York City (NYC), and Connecticut (CT) in the summer of 2023. We directly compare in situ vertical profiles to nearby Pandora partial columns and analyze day to day and spatial variability in Pandora columns. We also compare Pandora with

TEMPO and examine the influence of clouds on the relative agreement between the two measurements.

#### 2. Data/Methods

#### 2.1 AEROMMA

The NOAA airborne mission Atmospheric Emissions and Reactions Observed from Megacities to Marine Areas (AEROMMA; <a href="https://csl.noaa.gov/projects/aeromma/">https://csl.noaa.gov/projects/aeromma/</a>; last access July 2025) took place in the summer of 2023 with the goal of understanding urban and marine atmospheric composition. NASA's DC-8 aircraft was equipped with many in situ gas and aerosol sensors. Here we use observations from NASA's In Situ Airborne Formaldehyde instrument (ISAF; Cazorla et al., 2015) and NOAA's NO Laser-Induced Fluorescence instrument which measures NO, NO2, and NOy (NO-LIF; Rollins et al., 2020) where NOy is the sum of all reactive nitrogen oxides. Accuracies for HCHO and NO2 measurements are 15% and 10%, respectively. Potential temperature was measured with the Meteorological Measuring System (MMS). We focus primarily on four flights that took place over and around NYC on July 26th, July 28th, Aug 9th, and Aug 16th, 2023, hereafter referred to as "NYC1", "NYC2", "NYC3", and "NYC4". These flights and locations were chosen due to their proximity to Pandoras that perform MAX DOAS scans as discussed in the next section. In addition, we examine a single profile from the June 26th flight over southern CA.

### 2.2 Pandora Global Network

Pandoras operate in two modes: DS and MAX DOAS. During DS mode the lens of the Pandora follows the sun where the full width at half maximum field of view of the instrument is 2.5°. The fitting window for DS NO<sub>2</sub> and HCHO are 400-470 nm and 322.5-359.2 nm, respectively. The Pandora retrieves a slant column which, given a calculated geometric air mass factor (AMF), is converted into a total vertical column. Climatological stratospheric NO<sub>2</sub> is also provided in the DS data files based on latitude, season, and time of day (Brohede et al., 2007). The MAX DOAS product provides a 'tropospheric' column (integrating a vertical distance of ~3 km and a horizontal distance of ~10km), partial columns in different vertical layers, and a surface concentration using climatological air densities and differential AMFs. The surface concentration is estimated by either

the 90° solar zenith angle (SZA) scan or, more often, extrapolated from the largest zenith viewing angle measurement. The fitting windows for MAX DOAS NO<sub>2</sub> and HCHO are 435-490 nm and 328.5-359 nm, respectively. The MAX DOAS scans can be 'long' or 'short' in which long scans calculate 11 or 13 slant columns and short scans provide 4 columns (covering the same range of viewing angles). Further details on the algorithms used in the Pandora retrievals are available in the Blik user manual (Cede, 2024). We utilize level 2 (L2) vertical column data, in which spectral fitting has been applied, and columns have been calculated. Data are available at the NASA Pandora webpage (https://pandora.gsfc.nasa.gov/PGN/; Last access July 2025).

Table 1: Pandoras used in this study.

|                  |     | Latitude,           |             | Altitude Above | Pointing<br>Azimuth |
|------------------|-----|---------------------|-------------|----------------|---------------------|
| Pandora Name     | ID  | Longitude Longitude | Description | Sea Level (m)  | Angle (PAA)         |
| EdwardsCA        | 74  | 34.960, -117.881    | Southern CA | 692            | 30                  |
| BayonneNJ        | 38  | 40.670, -74.126     | NJ          | 3              | 316                 |
| BronxNY          | 180 | 40.868, -73.878     | NYC         | 31             | 250                 |
| MadisonCT        | 186 | 41.256, -72.553     | CT Coast    | 3              | 220                 |
| ManhattanNY-CCNY | 135 | 40.815, -73.951     | NYC         | 34             | 186                 |
| NewBrunswickNJ   | 69  | 40.462, -74.429     | NJ          | 19             | 45                  |
| NewHavenCT       | 64  | 41.301, -72.903     | CT Coast    | 4              | 300                 |
| OldFieldNY       | 51  | 40.963, -73.140     | Long Island | 3              | 38                  |
| QueensNY         | 55  | 40.736, -73.822     | NYC         | 25             | 330                 |
| WestportCT       | 177 | 41.118, -73.337     | CT Coast    | 4              | 218                 |
| NewLondonCT      | 236 | 41.376, -72.100     | CT Coast    | 30             | 90                  |

### 2.2.1 Pandora Data Quality Filtering

The PGN provides two sets of quality indicators for L2 data, each with three possible values for a total of nine combinations. First, the data are determined to be 'high', 'medium', or 'low' quality (denoted with a 0,1, or 2 in the ones place) based on an automated algorithm that flags exceedance of data quality thresholds (either instrumentation error or atmospheric sources). The Pandora documentation states that users should not use low quality data for most purposes (Cede, 2025).

The data are also assigned a modifier of 'assured', 'unassured', or 'unusable' (0,1, or 2 in the tens place), referring to the level of manual quality control. Once a PGN operator has inspected the data, they deem it 'assured' if there are no known issues in the quality assurance process or 'unusable' if there is a concern (often based on the spectral fitting or uncertainty). The rational for

data being labeled 'unusable' is not publicly documented with the dataset. 'Unassured' data has yet to be manually examined.

Fig. 1 shows the distribution of the L2 data quality flags for the 10 Pandoras used in this study (Table 1) from the start of their data collection (ranging from Sept 2019 to Sept 2022) through July 2025. As of July 2025, 52% of the data for all four products are 'unassured' due to a backlog for manual inspection. The distribution of high, medium, and low quality data varies between stations and data product. On average, 68% of the DS NO<sub>2</sub> data and 46% of MAX DOAS HCHO data are high quality. On the other hand, MAX DOAS NO<sub>2</sub> and DS HCHO are more often low quality (41% and 65%, respectively).

**Figure 1:** Distribution of the data quality flags for each station used in this study for (a) DS NO<sub>2</sub>, (b) DS HCHO, (c) MAX DOAS NO<sub>2</sub>, and (d) MAX DOAS HCHO. Gray bars indicate the portion of retained data for each station after applying the filtering method described in sect. 2.2.1.

The Pandora user manual recommends eliminating low quality data for most uses. However, due to the automated nature of flagging low quality data, there is a possibility of false negatives. *Rawat et al.*, *2024*, proposes an alternative empirical method to filter for poor data quality via four steps: 1) Determine a cutoff value for independent uncertainty given  $\mu$ +3 $\sigma$  (where  $\mu$  is the mean uncertainty and  $\sigma$  is standard deviation) for each Pandora based on the high quality data. 2) Filter the data to only include those beneath this threshold as well as data with a weighted room mean square error (wrms) < 0.01 and (for MAX DOAS only) a maximum horizontal distance < 20 km (range of which the MAX DOAS column spans in the horizontal direction). 3) Restore data with a percentage uncertainty under 10%. 4) Confirm based on the improved correlation between contemporaneous DS and MAX DOAS observations. For the present study, we use this proposed method and also require that the percent uncertainty be less than 30% for each scan. This additional criterion is particularly crucial for DS HCHO which has high uncertainty. For the monitors used in this study, removing only the low quality data retains an average of 65% of data, while the described filtering method retains an average of 82% (Fig. 1). SI text S1 further discusses the data filtering process.

## 2.2.2 Pandora Unit Conversion

Pandora partial columns are reported in moles/m<sup>2</sup> and must be converted into ppb for a direct comparison with airborne observations of HCHO and NO<sub>2</sub>. Climatological surface pressure and temperature (provided in the Pandora data files) are used to calculate air density, while the depth of the layer is assumed to be the distance between the layer heights given in the vertical profiles. The lowest level is assumed to extend to the surface. SI text S2 discusses this procedure in greater detail.

#### 2.3 *TEMPO*

Tropospheric Emissions Monitoring of Pollution (TEMPO) is the first geostationary satellite over North America to report column amounts of HCHO and NO<sub>2</sub> every 40-60 minutes at a resolution of 2.0 km N/S x 4.75 km E/W at the center of the field of regard. TEMPO was launched in April 2023 and provides data beginning August 2<sup>nd</sup> of the same year. The NO<sub>2</sub> retrieval is in the 405-465 nm spectral window, while HCHO retrieval is in the 328.5-356.5 nm fitting window

(Nowlan et al., 2025; Abad et al., 2025). These are slightly different ranges than the Pandora fitting windows for both DS and MAX DOAS modes. The validation plan for TEMPO includes diurnal and seasonal comparisons with airborne spectrometers, other satellites, and the PGN (Szykman, 2023; Szykman et al., 2025). We use version 3, level 2 HCHO data available from NASA's Earthdata website (Nasa, 2024; Last access July 2025). Using the nearest neighbor method, we grid L2 data to 2 x 4.5 km in the NYC domain bound by latitudes and longitudes of (40.2°, 41.5°) and (-74.6°, -71.7°), respectively. Data is screened to only include pixels with effective cloud fractions < 0.20 and data quality flags equal to 0. Daily means of the TEMPO HCHO column are calculated by averaging the HCHO values from each hour in all grid cells, and hourly domain averages are determined by averaging all valid data across the entire NYC domain for each hourly retrieval.

#### 3. Results

3.1 Case Study: Spiral over EdwardsCA Pandora

The DC-8 sampled in a vertical spiral over southern CA on June 26<sup>th</sup>, 2023. The spiral began at an altitude of 10.3 km above ground level (AGL) at 1:50 PM local time and ended at 0.03 km at 2:20 PM, including a low approach at Edwards Air Force Base (AFB). The flight track shown in Fig. 2 is centered over the EdwardsCA Pandora which, at this time, operated with a 30° pointing azimuth angle (PAA; red line). This is the only AEROMMA spiral that extended throughout the majority of the troposphere and took place directly over a Pandora monitor conducting MAX DOAS scans. This provides a rare opportunity for a direct comparison between Pandora vertical profiles and in situ observations.

The vertical profiles of in situ potential temperature, HCHO, and NO<sub>2</sub> are shown in Fig. 3 as the gray markers. Given the small variation in potential temperature at the lowest 2 km, followed by a sharp increase, the boundary layer (BL) appears to be well-mixed up to 1.97 km. In situ HCHO is also relatively constant in the BL (1.15  $\pm$  0.16 ppb;  $\mu$   $\pm$   $\sigma$ ) and followed by a sharp decrease in mixing ratio above the inversion (0.17  $\pm$  0.09 ppb).

**Figure 2.** Map of southern CA spiral colored by in situ (a) HCHO and (b) NO<sub>2</sub>. EdwardsCA Pandora (red triangle) and the viewing direction of Pandora MAX DOAS scans (red line) are included and extend to the mean integrated horizontal distance.

The in situ  $NO_2$  is more variable than HCHO (note the logarithmic x axis in Fig. 3c). Enhancements of  $NO_2$  up to 4.2 ppb are observed from 0.3 to 0.5 km in the southeast quadrant of the spiral. We define this enhancement in Fig. 3c as observations in the upper  $25^{th}$  percentile of data below 0.5 km (red triangles). Excluding this enhancement, the BL mean of  $NO_2$  is  $0.5 \pm 0.3$  ppb. There are no obvious local industrial sources (refinery, power plant, etc.) that would be a source of  $NO_2$  and there does not appear to be a consistent source of  $NO_2$  from any given wind direction based on historical Pandora data (SFig. S4). The  $NO_2$  enhancement is likely caused by local emissions from the nearby Edwards AFB airport (SFig. S5).

We include a total of 6 long scans and 16 short scans from the EdwardsCA Pandora that occurred within 1 hour of the spiral. The scans included are all 'high' quality data with average tropospheric HCHO and NO<sub>2</sub> column uncertainties of 9.0 x 10<sup>14</sup> molec/cm<sup>2</sup> (18%) and 8.0 x 10<sup>13</sup> molec/cm<sup>2</sup> (7%), respectively. Uncertainty is not reported for the partial columns, thus we include

the standard deviation of the points as the shaded region in Fig. 3 to represent the variability between each scan.

**Figure 3.** Vertical profiles of (a) potential temperature, (b) HCHO, and (c)  $NO_2$ . The boundary layer height is denoted as the dotted lines. The in situ airborne data are the gray markers and binned to 100 m (solid line). The shaded region represents measurement uncertainty. Pandora data are orange (HCHO) and blue ( $NO_2$ ) markers (solid lines are data binned to 100 m). The shaded region represents  $\pm$  one standard deviation. The red triangles in (c) are the  $NO_2$  enhancement observed in the southeast quadrant in Fig. 2, defined as the upper  $25^{th}$  percentile of the near surface observations (

222223

226227

229230

bias flips sign and Pandora HCHO are *low* by  $0.50 \pm 0.28$  ppb. Above the BL (1.97 km) there is comparatively closer agreement between the in situ and Pandora HCHO (a mean difference of 0.04 ppb). This result is consistent across the four different scan modes (SFig. S6).

Table 2 compares integrated vertical columns from the aircraft and Pandora. We vertically integrate the in situ observations to obtain two columns: one spanning the entire spiral (10 m to 10.3 km) and another limited to Pandora MAX DOAS tropospheric column (10 m to 2.5 km). The entire column is 6.6 x 10<sup>15</sup> molec/cm<sup>2</sup>, 25% lower than the mean total HCHO column found by the Pandora DS product during this time frame (8.8 x10<sup>15</sup> molec/cm<sup>2</sup>). Meanwhile, the average Pandora tropospheric HCHO column agrees with the integrated in situ column under 2.5 km to within uncertainty of the in situ data (5.2 x10<sup>15</sup> and 5.0 x10<sup>15</sup> molec/cm<sup>2</sup>, respectively). However, the individual partial columns have no reported uncertainty. The high bias near the surface and the low bias under the BL offset each other to produce a column amount of HCHO from the MAX DOAS scans that are very similar in value to the in-situ column.

Table 2: Column HCHO and NO2 during the southern CA spiral over EdwardsCA Pandora

HCHO column NO2 column (molec/cm<sup>2</sup>) (molec/cm<sup>2</sup>)  $6.6 \times 10^{15}$  $2.2 \times 10^{15}$ In Situ (0.01 - 10.3 km)  $5.0 \times 10^{15}$  $1.9 \times 10^{15}$ In Situ (0.01 - 2.5 km)  $1.9 \times 10^{15}$ Situ (0.01 - 10.3 km—no enhancement)  $1.6 \times 10^{15}$ Situ (0.01 -2.5 km-no enhancement) DS Total Columns (weighted mean)  $8.8 \times 10^{15} (\sigma = 8.9 \times 10^{14})$  $2.7 \times 10^{15} (\sigma = 2.5 \times 10^{14})$  $5.2 \times 10^{15} (\sigma = 8.7 \times 10^{14})$  $1.2 \times 10^{15} (\sigma = 1.4 \times 10^{14})$ MAX DOAS Tropospheric Column (weighted mean)

3.1.2 Pandora NO2

We compare the NO<sub>2</sub> profiles in Fig. 3c. For the entire profile, including all in situ observations, the Pandora NO<sub>2</sub> is 50% lower. Excluding near-surface enhancements reduces the

difference to 42% lower than the aircraft observations. The Pandora is pointing away from the surface enhancement and therefore is not expected to capture that feature. This is an important consideration for future comparisons to both in situ observations and high-resolution satellite retrievals. HCHO is sufficiently well-mixed in this case, such that the Pandora viewing angle does not appear to significantly affect the observed concentrations. This simply reflects a lack of strong local HCHO sources (most HCHO is produced through volatile organic compounds, VOC, oxidation at this time of year). NO<sub>2</sub>, on the other hand, can exhibit large horizontal and vertical gradients, especially in the presence of local emissions. Looking only at the lowest 0.5 km and excluding the enhancement, the Pandora is on average 15% greater than the in situ observations showing a similar pattern to the HCHO where the Pandora is greater at the surface. However, the turnover altitude for NO<sub>2</sub> is lower (0.3 km) compared to the HCHO profile (1 km).

While the partial columns are, on average, lower by 50%, this relationship is not reflected between the integrated columns and the DS total column or MAX DOAS tropospheric column. The mean Pandora tropospheric column NO<sub>2</sub> is 40% less than the integrated in situ column for lower than 2.5 km. Furthermore, the mean total DS NO<sub>2</sub> column (excluding the climatological stratospheric column) is 22% *greater* than the in situ integrated column for the entire spiral (2.7 x10<sup>15</sup> and 2.2 x10<sup>15</sup> molec/cm<sup>2</sup>, respectively). SFig. S2 shows the DS NO<sub>2</sub> is on average 52% greater than MAX DOAS NO<sub>2</sub> for EdwardsCA, one of the largest discrepancies for the 10 Pandoras in this study. It is possible that this low difference in MAX DOAS NO<sub>2</sub> is not a limitation of the Pandoras in general but rather an issue with this specific spectrometer.

To summarize, Pandora MAX DOAS HCHO agrees with the in situ observations with respect to the total tropospheric column, but the shape of the vertical profile exhibits a strong height-dependent bias. Quantification of the uncertainty in partial columns should be a priority, especially for columns near the tropopause. The Pandora MAX DOAS NO<sub>2</sub> is lower than the in situ observations for the entire profile, however better agreement exists at the surface between data within the viewing angle of the Pandora. This case study also emphasizes the need for consideration in the viewing angle of the Pandora during comparisons within horizontally heterogeneous environments.

### 3.2 New York City

The four NYC flights during AEROMMA passed over or near ten Pandora stations (Fig. 4). The density of the monitors allows us to evaluate the daily and spatial variability of Pandora HCHO. We use both the tropospheric columns derived from the MAX DOAS mode and the total columns from the DS scans. We do not analyze NO<sub>2</sub> because significant spatial heterogeneity in this domain degrades comparisons. Additionally, the PAA of several Pandoras are pointing away from the flight path (Table 1), making it difficult to determine whether aircraft-Pandora differences reflect instrument biases or urban NO<sub>2</sub> gradients.

**Figure 4.** Flight tracks for NYC1 (20230726), NYC2 (20230728), NYC3 (20230809), and NYC4 (20230816) colored by observed HCHO. Pandoras are marked as triangles. The altitude time series is included beneath the flight tracks colored by HCHO. The times and locations of vertical profiles discussed in sect. 3.2.2 are outlined in black.

Flights occurred within distinct meteorological and air quality conditions. NYC1 and NYC2 sampled under clear skies and warm surface temperatures (upper 80s to lower 90s F and a heat advisory during NYC2). Ozone exceedances occurred on both days, and the Air Quality Index (AQI) was labeled "unhealthy for sensitive groups." Mean observed HCHO was 2.3 ppb and 2.6 ppb for NYC1 and NYC2, respectively, with the highest mixing ratios over Long Island (LI) in the

midafternoon. Observed HCHO varied across the flight track these days (standard deviation 0.94 ppb and 0.70 ppb for NYC1 and NYC2, respectively), likely reflecting variability in both VOC emissions and oxidation rates.

NYC3 was slightly cooler (temperatures in the mid-80s) with moderate westerly winds and generally clear skies (with a few scattered cumulus in the afternoon). The BL is well mixed with moderate winds of 5-10 kts (10-20 m/s) near the surface (SFig. S7). HCHO mixing ratios were lower and less variable on this day, (mean 1.6 ppb, maximum of 2.6 ppb, standard deviation 0.3 ppb). Higher HCHO tends to be located over LI and downtown New York City.

There was scattered rain the night before and the morning of NYC4. The skies began to clear in the afternoon, but the domain remained partially cloudy all day, causing the DC-8 to take a different path than the previous flights. A stationary front to the south kept winds relatively calm (under 5 kts near the surface; SFig. S7). The mean observed HCHO remained relatively low (1.6 ppb) but with more variation than the previous flight (standard deviation 0.63 ppb). The maximum observed HCHO was 3.9 ppb with elevated levels occurring over Manhattan at 2:30 PM local time and over LI in the late afternoon. The AQI was 'good' to 'moderate' on both NYC3 and NYC4.

### 3.2.1 Pandora Tropospheric and Total Column Time Series

Figure 5 shows the 30 minute averaged time series of MAX DOAS tropospheric HCHO columns (Fig. 5a-d) and DS total HCHO columns (Fig. 5e-h) for the four NYC flight days. Individual stations (colored lines) and the average of all stations (gray dotted line) are included. Continuous data from Pandoras illustrate both the diurnal cycle and spatial variability of HCHO columns across the NJ-NYC-CT domain. These observations provide important insight into the temporal evolution of VOC–NOx interactions throughout the day (Sebol et al., 2024; Singh et al., 2024)

There is evident spatial variability in MAX DOAS HCHO on NYC1, NYC2, and NYC4. In contrast, the HCHO columns on NYC3 are lower and are more uniform across all monitors. This pattern aligns with the meteorological conditions on this day which featured moderately stronger horizontal winds and a deeper, well-mixed BL (SFig. S7).

Afternoon enhancements in tropospheric HCHO columns are observed on all days, however, the peak HCHO levels occur at different times across the domain. For example, on NYC1, BayonneNJ reaches its maximum HCHO at 1pm EDT while NewBrunswickNJ, 35 km away, continues to increase until 5pm EDT. WestportCT and OldFieldNY, both along the coastline of the Long Island Sound (LIS), remain relatively constant until approximately 2 pm EDT after which HCHO rapidly doubles. These observations highlight the ability of Pandoras to capture detailed temporal and spatial variations in HCHO, particularly near the land-ocean interface. This information complements "snapshot" aircraft measurements.

DS total column HCHO exhibits greater variability and more pronounced afternoon enhancements compared to MAX DOAS. On NYC1 and NYC2, MAX DOAS HCHO increased approximately 50% from morning to afternoon, while DS HCHO nearly doubled. Similarly, on NYC3, MAX DOAS HCHO doubled, but DS columns increased by 125%, with greater variability across stations. Despite clear-sky conditions, data gaps are still present in DS mode at certain sites, such as ManhattanNY-CCNY and MadisonCT, due to the higher sensitivity to atmospheric interference. As DS HCHO retrievals have higher uncertainty than MAX DOAS, more data are excluded during data filtering.

Limitations of both MAX DOAS and DS HCHO columns are apparent on NYC4, a mostly cloudy day. In the morning (8-12 EDT), when the clouds were most persistent, most data exceed the uncertainty threshold, however some MAX DOAS scans remain. After 12 EDT, once the clouds start to dissipate, all operational Pandoras produce valid MAX DOAS data revealing a strong east-west gradient in HCHO columns. In contrast, DS data were largely unavailable during this day, remaining sparse even after the clouds began to clear. The uncertainty in DS columns even during partial cloudiness makes data interpretation difficult.

These results align with the in situ airborne HCHO observations. Spatial variability was greater on NYC1 and NYC2 compared to NYC3, which had a low standard deviation ( $\sigma$  = 0.32 ppb) and a small range (~2 ppb for the flight). An east-west gradient was observed on NYC4, with values of the western being 2-3 times greater than those on the eastern side. From 12:30-18:30 local time (the earliest and latest times of flight for this domain), the mean in situ mixing ratios (averaged from the data shown in Fig. 4), MAX DOAS columns, and DS columns are 53% greater on NYC1 and NYC2 than on NYC3 and NYC4.

**Figure 5.** MAX DOAS tropospheric column (a-d) and DS total column (e-h) HCHO from NYC Pandoras on the four flight days averaged to 30 minutes. The individual stations are colored, and the mean value is shown as the dotted gray line. The hourly domain average TEMPO HCHO column is included on NYC3 and NYC4 as the solid black line.

The locations of the peak Pandora HCHO generally align with the highest in situ values. The largest in situ HCHO on NYC1 and NYC2 occurred over LI and downtown NYC around 15 and 16 EDT, respectively. These observations are consistent with elevated Pandora HCHO at similar times seen at OldFieldNY, WestportCT, and ManhattanNY-CCNY, indicating spatial consistency between the in situ and ground based spectrometers.

### 3.2.2 Vertical Profiles

While the DC-8 did not conduct high-altitude profiles over NYC as in the Edwards AFB case study, partial profiling was carried out on each flight allowing for comparisons with Pandora MAX DOAS vertical profiles. These times and locations are outlined in black on Fig. 4. Comparisons are limited to the Pandoras closest to each aircraft profile and those that had valid

long scans (11 or 13 levels). This is not ideal for evaluation of Pandora performance as differences may arise from air mass heterogeneity; however, it provides data for considering differences in vertical profiles. The DC-8 covered the approximate BL over the LIS, LI, and CT on NYC1 and NYC4 spanning from 0.11 to 1.92 km (NYC1) and 0.12 to 1.72 km (NYC4) and over a relatively large horizontal distance. The closest Pandoras to these profiles are OldFieldNY, WestportCT, and NewHavenCT. Spirals were done during NYC2 (from 0.13 to 3.02 km) and NYC3 (0.20 to 2.96 km) just west of the MadisonCT Pandora. Due to the height of surrounding buildings in NYC, surface concentrations from Pandoras may be limited.

Figure 6a compares partial columns from in situ and Pandora observations. Pandora vertical profiles (taken within  $\pm$  45 minutes of the aircraft profile time) are integrated over the altitude range covered by each aircraft profile. To highlight relative differences, columns are normalized to the in situ columns values on that day. Figure 6b shows the full vertical profiles.

The Pandora columns are greater than corresponding in situ columns (averaged over the same distance) primarily due to greater partial columns in the lower BL. On NYC1, WestportCT is 52% greater than the in situ column. The vertical profile at this site shows higher Pandora values from the lowest levels up to the BL height, where the difference reverses sign. On NYC4, NewHavenCT is 77% greater. In this case, Pandora values are greater both near the surface and above the BL. On this day, there was a lower, and less defined inversion. Neither the in situ HCHO nor these three Pandoras exhibit a distinct decrease at any altitude, rather they remain relatively constant from 0.12 to 1.72 km.

The only case when Pandora column HCHO is lower than the in situ occurs on NYC2 where OldFieldNY is 37% lower. This notable difference may be explained by the viewing geometry of the instrument. The Pandora uses a 38° PAA, facing the LIS, a region which is likely to exhibit lower HCHO concentrations compared to the areas sampled by the DC-8. This highlights how the distance and the heterogeneity of HCHO must be considered.

The most direct comparison to in situ data on these four days is on NYC3 with the NewHavenCT Pandora. The DC-8 completed a spiral just west of the Pandora and a single long scan was taken during the given time window. The integrated columns have near perfect agreement (both  $5.6 \times 10^{16} \text{ molec/m}^2$ ). Pandora HCHO is greater from the lowest level (0.45 km) to 1 km. From there, the Pandora becomes lower than the in situ profile.

**Figure 6**. (a) HCHO partial columns from integrated in situ observations during DC-8 profiles and mean Pandora MAX DOAS scans during the same time frame (± 45 minutes) from the nearest operational monitors normalized to the in situ column. Actual column values in molecules/cm² are included as the white text on the bars. We include 15% error (shaded red area) representing instrumentation accuracy of the in situ data; uncertainties for Pandora vertical profiles are not available. (b-e) Vertical profiles of HCHO from the NYC flights including the airborne in situ data (black solid line) and individual Pandora profiles (colored lines). Data are binned to 250m (except the NewHavenCT profile on NYC3). Dotted gray lines denote approximate boundary layer heights based on potential temperature gradients (not shown).

In an area such as New York City, where there are a variety of primary and secondary sources of HCHO and land/water meteorological influences there can be a horizontal gradient of HCHO that must be considered when interpreting Pandora columns. However, the Pandora MAX DOAS product does depict the day-to-day variability in total column amount, even while not capturing the precise profile shape, particularly in a deep, well-mixed BL. Future validation efforts would benefit from direct coordination between flight planners and the Pandora operators. Special attention in future validation efforts should be given to days with differing meteorological

conditions (BL height, wind patterns, clouds, etc.) to better understand the performance of the Pandora instruments.

# 3.2.3 TEMPO

Pandora column HCHO can be directly compared to TEMPO satellite retrievals during NYC3 and NYC4. Daily average TEMPO HCHO columns for these two days are shown in Fig. 7a and 7b. Just as in Pandora MAX DOAS columns and in situ observations, TEMPO HCHO is relatively uniform on NYC3, with only slightly elevated HCHO over downtown NYC and LI (mean  $9.6 \pm 2.4 \times 10^{15}$  molec/cm<sup>2</sup>), while on NYC4 there is greater mean HCHO column with an east-west gradient ( $12.8 \pm 5.0 \times 10^{15}$  molec/cm<sup>2</sup>).

**Figure 7**. (a,b) Daily average TEMPO HCHO columns for NYC3 and NYC4 from 8-17 EDT. (c,d) Number of data points included in the average for each grid box and time series of the percentage of valid data in the domain.

The number of valid pixels for each grid point and the total percentage of valid data for the domain are shown in Fig 7c and d. During NYC3, there was never less than 50% available data points until 17 EDT. In the late afternoon a few scattered cumulus clouds formed restricting some data along the CT coast. During NYC4, cloudiness reduced the fraction of valid pixels (> 75%)

until 1 pm EDT. After that, the skies cleared marginally, and valid data increased to nearly 50% for the rest of the day.

Mean hourly TEMPO HCHO columns are shown as the solid black line in Fig. 5c,d,g, and h. On NYC4 the mean value represents low-cloud areas only and thus may not be representative of the whole region. The agreement between average Pandora MAX DOAS HCHO columns and TEMPO HCHO columns is within 11% on NYC3. The standardized mean difference (SMD) takes the difference between two means and normalizes based on a weighted standard deviation of the data. In this case, SMD = 0.14, indicating a difference of less than one standard deviation. Both products show a similar afternoon increase in time and magnitude. On a clear-sky, well-mixed day, the ten Pandoras agree with the mean TEMPO retrievals for this NYC domain. In contrast, TEMPO HCHO is 30 - 40% lower than Pandora DS HCHO across the day of NYC (SMD = 1.3; over one standard deviation difference).

The agreement on NYC4 is influenced by the clouds. TEMPO reports twice the amount of HCHO as MAX DOAS HCHO from 8-13 EDT. However, starting around 14 EDT, the agreement between the two datasets improves, with a mean discrepancy of only 11% as data availability increases. At this time, valid TEMPO pixels never exceed 50%, but all 8 operational Pandoras (WestportCT and NewLondonCT were down on this day) have valid data. While TEMPO data lacks complete coverage under partially cloudy conditions, the MAX DOAS, which observes from the bottom-up, still provides valid data based on our filtering assumptions. DS HCHO is typically not valid under even partial cloud cover. Only BayonneNJ and OldFieldNY offer continuous DS HCHO.

To provide a more robust comparison, we extend the analysis for the entire month of August 2023. Figure 8 shows the correlation between mean hourly TEMPO HCHO and MAX DOAS HCHO (Fig. 7a) and DS HCHO (Fig. 7b) averaged over all stations from August 2nd to August 31st for the same NYC domain. For this analysis, we modify the Pandora filtering methodology by not removing data with greater than 30% uncertainty to reduce bias from eliminating low column amounts. We use the Theil Sen regression method where the slope is determined by finding the median of the slopes between all pairs of points (Ohlson and Kim, 2015) which reduces bias from outliers. Table 3 summarizes these statistics.

**Figure 8.** Average TEMPO HCHO column for NYC versus average NYC Pandora MAX DOAS tropospheric column (a) and Direct Sun total column (b) from August 2<sup>nd</sup> to August 31<sup>st</sup>, 2023. Markers are colored by the percentage of available TEMPO data for the domain. Data from NYC3 and NYC4 are outlined as the red X's and blue crosses, respectively. Theil Sen lines of best fit for all data (dotted) and those with > 20% valid data (solid) are included.

All three metrics comparing TEMPO and MAX DOAS HCHO (slope, y-intercept, and R<sup>2</sup>) improve by removing the cloudiest points or when the fraction of valid TEMPO data across the domain is less than 20%. The cutoff value of 20% was chosen to provide the best correlation while retaining the most data points (SFig. S8). On average, TEMPO HCHO is greater than MAX DOAS HCHO at higher concentrations. The differences between NYC3 and NYC4 are apparent where NYC3 follows the pattern of the clear sky data points and NYC4 has very little correlation to TEMPO. Similarly, removing the cloudiest of data points improves all three metrics for the TEMPO HCHO and DS HCHO comparison. The slope of 0.93 and a higher y intercept of 3.3 x 10<sup>15</sup> molec/cm<sup>2</sup> indicates a systematic difference between TEMPO HCHO and DS HCHO, not dependent on the magnitude of column HCHO. These results suggest an unidentified source of bias in TEMPO HCHO. According to a priori HCHO profiles provided with the TEMPO retrieval, approximately10-15% of the HCHO column should reside above the ~3 km ceiling of the Pandora MAX DOAS retrieval.

**Table 3:** Theil Sen regression parameters (slope, y-intercept, and R<sup>2</sup>) corresponding to the data shown in Fig. 8, representing the relationship between TEMPO HCHO and Pandora HCHO. >20% refers to the removal of data points when the fraction of valid TEMPO data across the domain is less than 20%.

|                          | TEMPO v              | MAX DO               | AS                   |                      | TEMPO v Direct Sun   |                      |                      |                      |
|--------------------------|----------------------|----------------------|----------------------|----------------------|----------------------|----------------------|----------------------|----------------------|
|                          | All                  | > 20%                | NYC3                 | NYC4                 | All                  | > 20%                | NYC3                 | NYC4                 |
| Slope                    | 0.63                 | 0.85                 | 0.80                 | -1.4                 | 0.75                 | 0.97                 | 0.94                 | -1.4                 |
| Y-Intercept              | 1.2x10 <sup>15</sup> | $8.0 \times 10^{13}$ | 9.2x10 <sup>14</sup> | 2.5x10 <sup>16</sup> | 3.6x10 <sup>15</sup> | 2.0x10 <sup>15</sup> | 3.9x10 <sup>15</sup> | 3.1x10 <sup>16</sup> |
| (molec/cm <sup>2</sup> ) | 1.2/10               | 0.0210               | J.2K10               | 2.5810               | J.OATO               | 2.0110               | 3.7A10               | 3.1A10               |
| $\mathbb{R}^2$           | 0.24                 | 0.66                 | 0.99                 | 0.25                 | 0.32                 | 0.68                 | 0.94                 | 0.35                 |

### 4. Conclusions

The PGN provides long term, continuous trends of column HCHO and NO<sub>2</sub>, filling observational gaps in airborne observations and providing information under clouds where TEMPO cannot see. It is imperative to understand the strengths and limitations of the PGN, especially as it is a key component to the TEMPO validation plan and the HCHO products are relatively new and unvalidated themselves.

This evaluation includes a vertical spiral conducted by the DC-8 over the EdwardsCA Pandora monitor in southern CA and four flights that took place in and around New York City. We find that integrated HCHO columns are consistent between Pandoras and in situ airborne measurements. However, the shape of the Pandora HCHO profile is biased high near the surface (lower 0.5 km) and then biased low beneath the BL under well mixed conditions as shown in the EdwardsCA case study. This pattern of disagreement also manifests on a well-mixed day in NYC. Work is ongoing in the Pandora group to isolate and correct this artifact.

Due to the greater spatial heterogeneity of NO<sub>2</sub>, particularly in urban areas, the comparison between Pandora and other data must be taken with extra care to ensure the same space is measured. The EdwardsCA spiral showed a significant surface enhancement outside the viewing angle of the Pandora but within the horizontal range of the slant columns. Without consideration of the viewing angle, the MAX DOAS NO<sub>2</sub> in the bottom 0.5 km is 27% *lower* than the in situ

tropospheric column and 15% *greater* after removing the enhancement showing a similar altitude discrepancy as in HCHO. However, above this enhancement, the Pandora NO<sub>2</sub> vertical profile is on average 55% lower than the in situ observations. This issue is likely relevant for comparison of Pandora and TEMPO NO<sub>2</sub>, especially in urban areas. To mitigate such issues, future validation of TEMPO and Pandora NO<sub>2</sub> should focus on more homogeneous regions, such as suburban and rural areas.

The day-to-day variability in Pandora HCHO over NYC aligns with the in situ HCHO trends, where the greatest values are observed on the first two flight days. Collectively, the ten Pandoras deployed across NYC effectively represent the relative distribution of HCHO offering context for interpreting air quality conditions and ozone exceedances. Pandoras also provide detailed information regarding spatial and diurnal variations in HCHO, such as differences between urban and coastal sites. The DS HCHO appears to be more sensitive to data availability, particularly under cloudy conditions whereas MAX DOAS HCHO exhibits lower uncertainty.

We also compared TEMPO HCHO retrievals to Pandora columns during the AEROMMA time frame. According to the current TEMPO version 3 user manual, data with a cloud fraction above 0.2 should not be used for analysis— a threshold that is more conservative than the 0.3-0.4 typically used for other satellites. As a result, data availability is limited under even moderately cloudy conditions. However, there are still continuous MAX DOAS HCHO data under partially cloudy conditions which, being observed from the ground up and through differential slant columns, is not as sensitive to the presence of clouds. When eliminating the cloudy hours (data availability 

539540

must include long scans within 30 minutes of the aircraft profile. The PAA of the Pandoras is semifixed and usually chosen to avoid tall buildings or trees that would cause interference. Because of this the viewing angle of the Pandora is not always easy to change. Flight planners for future missions should be aware which angle the Pandoras are pointing to collect data for accurate comparisons, particularly regarding NO<sub>2</sub>. In the same vein, Pandora site selection might include consideration of air traffic restrictions. Balloon or drone-borne measurements may provide an alternative for less costly and more frequent vertical profile comparison (Bailey et al., 2024).

A clearer understanding of the Pandora data quality filtering process and quantitative uncertainties for all data products are also necessary for the broader research community. Currently, there are no official materials offering guidance on this procedure, largely due to the limited number of formal validation studies particularly on the MAX DOAS HCHO and NO<sub>2</sub> and the DS HCHO products. Robust uncertainty estimates and transparent quality assurance procedures will enhance the scientific utility of these datasets.

### **Code Availability**

Code Available upon request

#### 545 Data Availability

- All data are publicly available online at the following locations:
- <a href="https://csl.noaa.gov/projects/aeromma/">https://csl.noaa.gov/projects/aeromma/</a> (AEROMMA)
- <a href="http://data.pandonia-global-network.org">http://data.pandonia-global-network.org</a> (Pandora Global Network)
- <a href="https://search.earthdata.nasa.gov/">https://search.earthdata.nasa.gov/</a> (TEMPO)

#### **Author Contribution**

- AS prepared the manuscript with contributions from all coauthors. AS, GW, JSC, ED, JK, and ND
- collected ISAF data. DR, EW, and KZ collected NOy-LIF data. CG and JDD collected MMS data.
- BP and AP provided their expertise on the Pandora Network. AS provided guidance on using
- TEMPO data. GW, TC, AR contributed valuable suggestions on analysis and conclusions.

571

578

579

### **Competing Interests**

558 GW is a member of the editorial board of AMT.

## Acknowledgements

- Authors thank the entire AEROMMA science team for making the airborne data collection possible
- through their research, flight preparations, and measurement collections. We also express our
- gratitude towards the Pandora team at LuftBlick, SciGlob, and NASA Goddard Space Flight
- Center (including Tom Hanisco) for their work in providing the Pandora data. Caroline Nowlan
- and Gonzalo González Abad contributed valuable advice in handling TEMPO data. Finally, we
- would like to recognize the EPA and PIs of all the Pandora instruments used in this study that have
- worked to provide data: James Podolske, Nader Abuhassan, Lukas Valin, Maria Tzortziou, James
- Szykman, Eric Baumann and Brooke Stutzman. CSL authors gratefully acknowledges the
- generous support of the NOAA NESDIS GeoXO Program, which enabled the use of the NASA
- DC-8 aircraft and associated costs for NASA flight crew, engineering, integration, and logistics
- for the AEROMMA mission.

### 572 Financial Support

- This work was supported by the following: National Science Foundation (NSF; Grant # AGS-
- 2023605), the NASA Tropospheric Composition program, Atmospheric Composition Campaign
- Data Analysis and Modeling (ACCDAM; Grant # 80NSSC21K1448), NASA UACO
- (NNH20ZDA001N-UACO), NOAA NA21OAR4310137, and NOAA Cooperative Agreement
- (NA22OAR4320151).

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
