# Peer review of "Evaluation of Pandora HCHO and NO2 with Airborne In Situ Observations 1"

_EGUsphere, 2025_

## Referee Comment (RC2)

Review of "Evaluation of Pandora HCHO and NO$_2$ with Airborne In Situ Observations"

This manuscript presents a valuable evaluation of Pandora HCHO and NO$_2$ profile measurements using airborne in situ observations. The study is well designed, the analysis is generally robust, and the results are relevant for the validation of ground-based and satellite air quality products. Overall, the paper makes a meaningful contribution.

The conclusions are clearly written and appropriately summarize the key findings, and the discussion of the New York flight conditions is particularly clear and well presented. I especially appreciate that the authors highlight the importance of coordinated aircraft and Pandora observations in future work. I recommend this manuscript for publication after addressing the comments and minor revisions outlined below.

Below are specific comments and suggestions intended to improve clarity.

General and Section-Specific Comments
1. The conclusion that differences in NO$_2$ are primarily driven by spatial heterogeneity, while HCHO profiles show better agreement, is not strongly supported by Figure 3. As stated in the abstract, MAX-DOAS NO$_2$ is reported to be ~80% lower for the entire profile and particularly sensitive to the Pandora viewing angle due to NO$_2$ spatial heterogeneity. To better support this conclusion, it would be valuable to demonstrate whether the observed biases decrease when the aircraft data are restricted to sampling along (or close to) the Pandora pointing azimuth. In addition, it would be helpful to clarify whether spatial heterogeneity varies with altitude (e.g., being stronger near the surface and weaker aloft).

2. **Introduction**:
   The introduction is concise and effective. However, adding a few lines describing the geographical regions studied (e.g., New York, Long Island Sound, Edwards CA) would help orient readers early in the manuscript.

3. **Line39**:
   Please include SO$_2$ and water vapor columns among the Pandora products, and clarify that total ozone is retrieved (i.e., no partial ozone product is available).

4. **Table1**:
   In the table caption, please clarify that the listed pointing azimuth angles correspond to sky-scan (MAX-DOAS) observations.

5. **Line218**:
   Please explicitly refer to Figure 3 when discussing these results.

6. **Line221/Section2.2**:
   Please add a brief explanation of the "no filter" vs. "yes filter" cases to help readers interpret Figure S6.

7. **Lines 223–224**:
   Please confirm that the 10 m altitude refers to above ground level (AGL). Also discuss whether the missing column from the surface to 10 m could contribute to the observed differences between Pandora and aircraft columns.

8. **Tables (general)**:
   Please expand table captions to be more descriptive, making them more self-contained for readers.

Interpretation and Methodology Clarifications

1. **Line 248**:
   The higher near-surface Pandora columns for both $NO_2$ and HCHO warrant further discussion. Could this be related to the aircraft spiral sampling a broader horizontal area? Please consider including the range of spiral radii at different altitude.
2. **Line 250**:
   The term "integrated column" for MAX-DOAS needs clarification. Are the authors integrating profile layers themselves, or are they using independently retrieved tropospheric columns from sky-scan/MAX-DOAS? Typically, MAX-DOAS columns are retrieved directly and not calculated by vertically integrating profiles. Please clarify and revise wording if needed.
3. **Line 255**:
   Could the DS $NO_2$ measurements be influenced by **airport emissions** during morning observations due to pointing azimuth angles? It would be useful to examine how the difference between DS and SS $NO_2$ columns varies with **solar azimuth - pointing azimuth**.
4. **Lines 260–261**:
   I doubt that MAX-DOAS has sensitivity up to tropopause altitudes. Please clarify or revise this statement.
5. **Line 263**:
   I may be missing, but I do not see a clear comparison of $NO_2$ MAX-DOAS profiles or columns restricted to aircraft sampling along (or near) the pointing azimuth direction. If this analysis is included under the "no enhancements" category in Table 2, please explicitly mention and cite it.
6. **Line 283**:
   Please clarify whether the reported HCHO values are based on aircraft data, and/or cite Figure 4**.**
7. **Line 326**:
   Rawat et al. (2025) reported issues with Manhattan observations. Similar behavior appears evident here. I recommend adding a brief statement acknowledging known Pandora DS limitations at this site.
8. **Line 360**:
   There is some confusion here. Even though the Edwards CA aircraft profile extends from 0.1 to 10 km, MAX-DOAS sensitivity is typically limited to ~2–3 km. Please clarify why the New York spiral (0.1–2 km) is considered less ideal in this context.
9. **Line 375**:
   Please note that the 77% in situ–Pandora difference at New Haven, CT may also be influenced by the limited vertical representativeness of the aircraft profile.
10. **Line 381**:
    The Long Island Sound (LIS) region frequently experiences ozone exceedance events and is often downwind of major emission sources. One might therefore expect higher HCHO.

Could you justify the lower HCHO values over LIS using aircraft data and, if possible, please provide values?

Figures and Spatial Interpretation

1. **Section 3.2.3**:
   Please explain why only TEMPO HCHO is evaluated in this section. Alternatively, I encourage the authors to conduct a similar TEMPO assessment for $NO_2$, comparable to the thorough HCHO evaluation presented. Similarly, why no comparison of TEMPO columns with aircraft observations.

2. **Lines 427–445 / Figure 5**:
   Please clarify whether the TEMPO diurnal profiles are derived from:
   a. all pixels within the New York domain, or
   b. first collocated with Pandora and then averaged.

   Revising this section for clarity would greatly improve interpretation.

3. **Figure 8**:
   Please describe the spatial domain and collocation criteria used. Also discuss how the choice of domain may influence the observed biases.